# Privacy Assessment on Reconstructed Images: Are Existing Evaluation Metrics Faithful to Human Perception?

**Xiaoxiao Sun**[†]
Australian National University
xiaoxiao.sun@anu.edu.au

**Nidham Gazagnadou**[‡]
Sony AI
nidham.gazagnadou@sony.com

**Vivek Sharma**[‡]
Sony AI
viveksharma@sony.com

**Lingjuan Lyu**[‡ ✉]
Sony AI
lingjuan.lv@sony.com

**Hongdong Li**[†]
Australian National University
hongdong.li@anu.edu.au

**Liang Zheng**[†]
Australian National University
liang.zheng@anu.edu.au

## Abstract

Hand-crafted image quality metrics, such as PSNR and SSIM, are commonly used to evaluate model privacy risk under reconstruction attacks. Under these metrics, reconstructed images that are determined to resemble the original one generally indicate more privacy leakage. Images determined as overall dissimilar, on the other hand, indicate higher robustness against attack. However, there is no guarantee that these metrics well reflect human opinions, which offers trustworthy judgement for model privacy leakage. In this paper, we comprehensively study the faithfulness of these hand-crafted metrics to human perception of privacy information from the reconstructed images. On 5 datasets ranging from natural images, faces, to fine-grained classes, we use 4 existing attack methods to reconstruct images from many different classification models and, for each reconstructed image, we ask multiple human annotators to assess whether this image is recognizable. Our studies reveal that the hand-crafted metrics only have a weak correlation with the human evaluation of privacy leakage and that even these metrics themselves often contradict each other. These observations suggest risks of current metrics in the community. To address this potential risk, we propose a learning-based measure called **SemSim** to evaluate the **Sem**antic **Sim**ilarity between the original and reconstructed images. SemSim is trained with a standard triplet loss, using an original image as an anchor, one of its recognizable reconstructed images as a positive sample, and an unrecognizable one as a negative. By training on human annotations, SemSim exhibits a greater reflection of privacy leakage on the semantic level. We show that SemSim has a significantly higher correlation with human judgment compared with existing metrics. Moreover, this strong correlation generalizes to unseen datasets, models and attack methods. We envision this work as a milestone for image quality evaluation closer to the human level. The project webpage can be accessed at https://sites.google.com/view/semsim.

## 1 Introduction

This paper studies the *evaluation* of privacy risks of image classification models, with a focus on reconstruction attacks [6, 43]. During inference, a target classifier, a reconstruction attack algorithm

37th Conference on Neural Information Processing Systems (NeurIPS 2023).

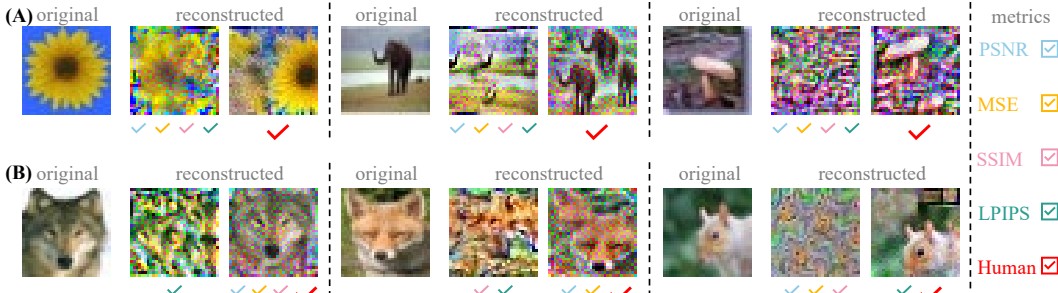

Figure 1: **Inconsistency between existing metrics and human judgements on privacy information leakage.** For each original image, we present two reconstructions produced by InvGrad [8]. Below the reconstructed images, each colored ✓corresponds to a different metric, indicating that the corresponding metric evaluates the reconstruction to have more information leakage. In **(A)**, according to PSNR, MSE, SSIM and LPIPS, the first reconstructed image is evaluated to have ***more*** privacy leakage [8, 7] than the second one (*i.e.*, the first one has a higher PSNR, SSIM values, and a lower MSE and LPIPS values). However, human annotators perceive the first image as having ***less*** privacy leakage, since they cannot recognise this recognition (in contrast to the second reconstruction, which is recognizable and suggested to have more information leakage). *Such inconsistency in privacy assessment is our key observation and motivation.* Moreover, we observe in **(B)** that even these metrics themselves often disagree with each other.

and a test set are used. For each original test image, the attack algorithm intercepts gradients of the target model to obtain a reconstructed image [7, 40]. The evaluation objective is to measure whether the reconstructed image leaks any private information of the original one.

In the literature, *objective evaluation metrics* [29, 25] such as peak signal-to-noise ratio (PSNR), mean squared error (MSE) and structural similarity index (SSIM) are commonly used. They measure the similarity between two images on the pixel-level. In common practice, the high similarity between the original and reconstructed image indicates a good reconstruction attack, thus a more vulnerable classification model. Conversely, the low similarity between the two images means poor reconstruction, which is believed to indicate weak privacy risk.

However, it is often subject to *human perception* whether privacy is leaked or preserved. In Figure 1, we show examples where hand-craft evaluation metrics, such as PSNR and SSIM, and CNN-feature-based metric learned perceptual image patch similarity (LPIPS) [39] give different judgments of privacy assessment on reconstructed images from human perception. For example, in **(A)**, the reconstructed image that is recognizable (privacy-leaked) by annotators is evaluated as better privacy-preserved by PSNR, SSIM, and LPIPS. In **(B)**, sometimes some of these metrics provide consistent judgments with human annotators, but their evaluation accuracy is still unstable for different images.

In light of the above discussions, this paper raises the question: *is model privacy preservation ability as measured by existing metrics faithful to human perception?* To answer this question, we conduct extensive experiments to study the correlation of model privacy preserving ability measured by human perception and existing evaluation metrics. Specifically, for each reconstructed image, we ask five independent annotators whether the reconstruction is recognizable. We use the average annotator responses over the test set as human perception of privacy information leakage. On a wide range of scenarios (5 datasets of different concepts, many different classification models and 4 reconstruction attack methods), we find that there is only a weak correlation between human perception and existing metrics. It suggests that a model determined as less vulnerable to reconstruction attacks by existing metrics may actually reveal more private information as judged by humans.

Recognizing such discrepancy, we propose a new learning-based metric, semantic similarity (Sem-Sim), to measure model vulnerability to reconstruction attack. Using binary human labels that indicate whether a reconstructed image is recognizable, we train a simple neural network with a standard triplet loss function. For an unseen pair of images, we extract their features from the neural network and compute their $\ell_2$ distance, which is referred to as the SemSim score. If a model has a low (resp. high) average SemSim score, it is considered to have a high (resp. low) risk of privacy leakage, We experimentally show models' vulnerability to reconstruction attack which is ranked by SemSim has a much stronger correlation with human perception than existing metrics. Our main contributions are summarized below.

- We find model privacy leakage against reconstruction attacks measured by existing metrics is often inconsistent with human perception.

- We propose SemSim, a learning-based and generalizable metric to assess model vulnerability to reconstruction attack. Its strong correlation with human perception under various datasets, classifiers and attack methods demonstrates its effectiveness.

- We collect human perception annotations on whether privacy is preserved for 5 datasets, 14 different architectures of each set, and 4 reconstruction methods. These annotations will become valuable benchmarks for future study.

## 2 Related Work

**Image quality and similarity metrics** are usually used to indicate the performance of reconstruction attack approaches [42, 43, 41] and also in privacy assessment [7, 36, 32] of methods against reconstruction attacks. These metrics can be broadly categorized into pixel-level and perceptual metrics. Pixel-level metrics, such as PSNR [14, 29] and MSE [34], evaluate differences between pixel values of the original and reconstructed images [7, 36, 32], to reflect the degree of privacy leakage. Perceptual metrics, such as SSIM [35] and LPIPS [39] are designed to take into account the perceptual quality of images for privacy leakage evaluation [13]. This paper examines the effectiveness of these metrics in privacy leakage evaluation and finds they exhibit weak correlation with human annotations.

**Reconstruction attacks** [43, 8, 41, 42, 1] aim to recover the training samples from the shared gradients. Phong *et al*. [26] show provable reconstruction feasibility on a single neuron or single layer networks, which provide theoretical insights into this task. Wang *et al*. [33] propose an empirical approach to extract single image representations by inverting the gradients of a 4-layer network. Meanwhile, Zhu *et al*. [43] formulate this attack as an optimization process in which the adversarial participant searches for optimal samples in the input space that can best match the gradients. They employed the L-BFGS [19] algorithm to implement this attack. Zhao *et al*. [41] extend the approach with a label restoration step, hence improving speed of single image reconstruction. We focus on model privacy assessment against reconstruction attacks and evaluate different metrics using several attack methods.

**Human perception annotations** play an essential role in evaluating machine learning models [22, 20, 27, 38]. Most public test sets, such as the ImageNet [2] dataset from the computer vision, are annotated by humans, allowing for conventional evaluation. Moreover, human feedback has been used to improve machine learning models, such as InstructGPT [24]. In fields where human annotations were expensive to obtain, *e.g.*, medical image analysis [37] and image generation [28], there is increasing evidence that the human judgements or evaluation is valuable and offers new insights. In our paper, we consider the information leakage of reconstructed images

## 3 Privacy Assessment Metrics on Reconstructed Images: A Revisit

**Pipeline of privacy assessment on the reconstructed images.** As shown in Figure 2, the goal of evaluation is to compare privacy risks of a series of $K$ image classification models $\{\mathcal{M}_k\}_{k=1}^K$, under reconstruction attacks. The evaluation process simulates stealing data from gradients [43, 41]. Its input consists of an original image set $\mathcal{X} = \{\mathbf{x}_i \in \mathbb{R}^{m \times n}\}_{i=1}^N$, where $N$ is the number of images, and a reconstruction algorithm $\mathcal{A}$ used to attack models $\{\mathcal{M}_k\}_{k=1}^K$. Given a target model $\mathcal{M}^1$, whose parameter weights are denoted by $\mathcal{W}$, its gradients $\nabla \mathcal{W}_{\mathcal{X}}$ can be calculated using the original data $\mathcal{X}$. The attack algorithm $\mathcal{A}$ is applied to the target model $\mathcal{M}$ and its gradients $\nabla \mathcal{W}_{\mathcal{X}}$ to obtain a set of reconstructed images denoted by $\bar{\mathcal{X}} := \mathcal{A}(\mathcal{M}, \nabla \mathcal{W}_{\mathcal{X}}) = \{\bar{\mathbf{x}}_i\}_{i=1}^N$. Note that, $\mathcal{A}$ can access the gradients, but has not access to $\mathcal{X}$. We can evaluate the privacy leakage of a target model $\mathcal{M}$ over the original set of $\bar{\mathcal{X}}$ as follows:

$$\mathrm{PL}(\mathcal{M}) := \mathrm{InfoLeak}(\mathcal{X}, \bar{\mathcal{X}}) = \mathrm{InfoLeak}(\mathcal{X}, \mathcal{A}(\mathcal{M}, \nabla \mathcal{W}_{\mathcal{X}})), \tag{1}$$

where $\mathrm{InfoLeak}(\cdot, \cdot)$ represents the amount of information leakage in reconstructed images. Therefore, it is important to have an effective metric for indicating $\mathrm{InfoLeak}(\cdot, \cdot)$.

---

[1]Unless explicitly stated otherwise, the subscript of $\mathcal{M}$ is omitted when this does not create ambiguity.

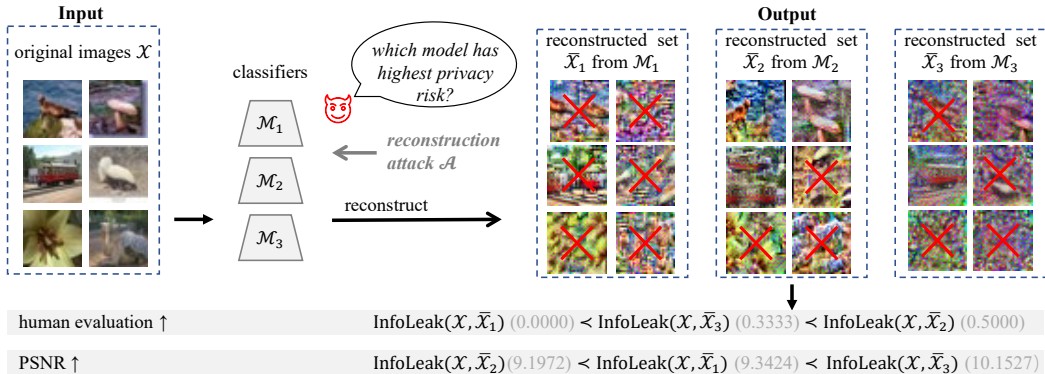

Figure 2: **Task definition: privacy leakage assessment on reconstructed images.** Given $K$ classification models $\mathcal{M}_1, \mathcal{M}_2, ..., \mathcal{M}_K$ against the image reconstruction attack $\mathcal{A}$ (we use $K = 3$ as an example in this figure) on a set of original images $\mathcal{X}$. For each model, we get a set of reconstructed images. The main goal of privacy leakage assessment on reconstructed images is to measure whether semantic information of an original image, is still accessible. We can ask human annotators to evaluate whether they can recognize the image class and then average across the set of images to obtain the overall human evaluation score of privacy leakage. In the existing literature, image quality metrics, such as PSNR, are used to measure privacy leakage. Here, the evaluation of example images shows again that PSNR deviates from human evaluation.

**Information leakage formulation.** As introduced in Section 2, information leakage is often as-similated to reconstruction quality and is based on a distance between an original image $\mathbf{x}_i$ and its reconstructed counterpart $\bar{\mathbf{x}}_i$. Under such pointwise metric, InfoLeak$(\cdot, \cdot)$ of an image set $\mathcal{X}$ and its reconstructed set $\bar{\mathcal{X}}$ can be defined as:

$$\text{InfoLeak}(\mathcal{X}, \bar{\mathcal{X}}) = \frac{1}{N} \sum_{i=1}^{N} d(\mathbf{x}_i, \bar{\mathbf{x}}_i), \tag{2}$$

where $d$ can be a hand-crafted metric, such as MSE [34], PSNR [14, 29] or SSIM [35], or model based, such as LPIPS [39]. Equation (2) averages the distances or similarities over all the original - reconstructed image pairs to obtain the information leakage score of the attacked model $\mathcal{M}$. Apart from these, we can also use Fréchet Inception Distance (FID) [10]. It measures information leakage as the distribution difference between original and reconstructed images: InfoLeak$(\mathcal{X}, \bar{\mathcal{X}}) \propto \text{FID}(\mathcal{X}, \bar{\mathcal{X}})$.

## 4 Diagnosis of Existing Metrics and Our Proposal

### 4.1 Collecting human assessment of privacy leakage from reconstructed images

To evaluate whether a reconstructed image leaks privacy, human perception offers very useful judgement. In the context of image recognition and face recognition, it is to determine if the human can still recognize the reconstructed object or face.

For **image classification**, given an image, we provide human annotators with an incomplete list of classes. For example, for the CIFAR-100 dataset, instead of providing annotators with a list of all the 100 classes which are hard to memorize, we provide them with a list of the top-20 possible classes that includes the ground truth. We request annotators to annotate the class of a given image. If the annotate thinks the images is "incomprehensible" (*i.e.*, severely blurry) or the right class does not appear in the candidate list, then the annotation is 'none'. We compare the human annotations between an image and its reconstructed version. If they are the same, privacy is not preserved; otherwise, privacy is preserved. The annotation pipeline and more details of the annotation process are provided in the supplementary material.

For **face recognition and fine-grained image recognition**, because it is by nature very difficult for a human to assign a class label from 20 candidates, we give annotators two images at a time: an original image and its reconstruction. We then ask the annotator to tell whether the two images

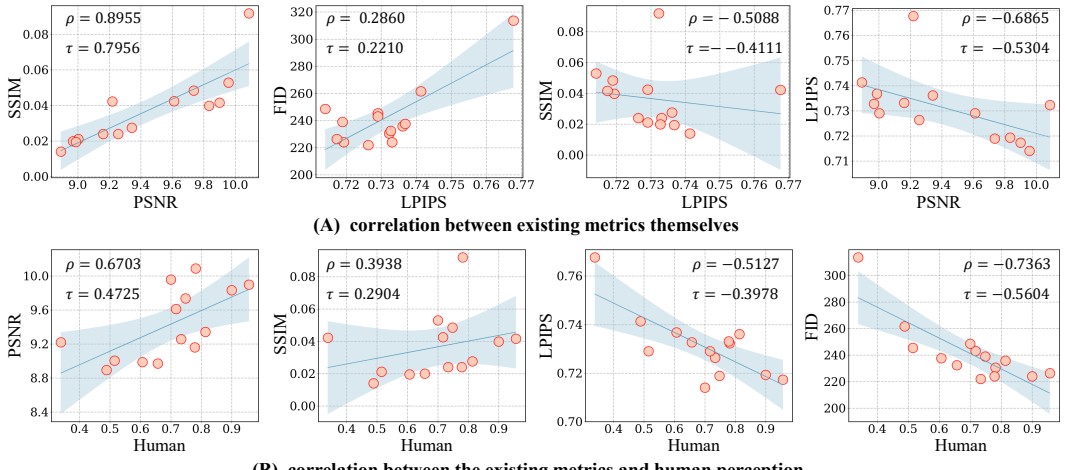

Figure 3: **Correlation between existing metrics and their alignment with human perception in measuring privacy risk.** 14 classification models are attacked by InvGrad [8] on the CIFAR-100 dataset. Each subfigure presents the correlation between the rankings of model privacy leakage obtained by two metrics. The correlation strength is measured by Spearman's rank correlation ($\rho$) [31] and Kendall's rank correlation ($\tau$) [16]. Between existing metrics, **(A)** indicates that correlation is sometimes very weak. Furthermore, **(B)** indicates that the correlation between existing metrics and human perception is generally weak.

contain the same person or category. If yes, then privacy is not considered as preserved; otherwise, it is. Note that in this procedure, to mitigate the potential bias of annotators, we also give reconstructed images that do not pair with the original image.

In all the above procedures, each image or image pair is labeled by 5 independent annotators. Binary labels, *i.e.*, whether a reconstructed image is recognizable, are obtained via majority voting. In this study, we deal with five datasets: CIFAR-100, Caltech-101, Imagenette and Celeb-A and Stanford Dogs[2]. For each classification model being attacked, we annotate 600, 700, and 100 reconstructed images for the CIFAR-100, Caltech-101, and the other three datasets, respectively.

### 4.2 Correlation analysis between human perception and existing metrics

Examples from Figure 1 motivate us to conduct a more comprehensive analysis of the inconsistency between human perception and existing metrics in terms of privacy leakage. To this end, for the reconstructed image set of 14 target models, we plot their privacy risk measured by various metrics against collected human labels in Figure 3 **(B)**. We find that the correlation strength between human evaluation and existing metrics is relatively weak. For example, Kendall's rank correlation $\tau$ that measures rank consistency is only 0.2904 and 0.3978 for SSIM and LPIPS. Even in the best case, *i.e.*, FID *vs* human, correlation is only moderate with $\tau = 0.5604$. It signifies that a model identified as more robust against reconstruction attacks based on existing metrics may actually be perceived as highly vulnerable according to human judgment when comparing different models.

The primary issue lies in the fact that existing metrics are computed on either a pixel-wise or patch-wise basis, without considering the semantic understanding of privacy leakage. As a result, these metrics fail to accurately capture the image semantics related to privacy risks. This problem motivates us to design privacy-oriented metrics to better assess privacy leakage.

### 4.3 Proposed metric

To obtain a metric that is more faithful to human perception, we propose SemSim, a learning-based metric using human annotations as training data. The pipeline of SemSim is presented in Figure 4.

**Training.** Using binary human labels whether a reconstructed image is recognizable, we train a simple neural network $f_{\boldsymbol{\theta}}$ with a standard triplet loss function. We take the original image $\mathbf{x}_i$ as an anchor

---

[2]The new annotated dataset is distributed under license CC BY-NC 4.0 1, which allows others to share, adapt, and build upon the dataset and restricts its use for non-commercial purposes.

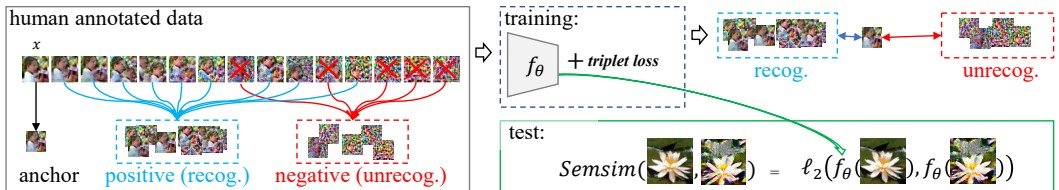

Figure 4: **Training and inference pipeline of SemSim.** Feature extractor $f_\theta$ is trained on human-annotated images with a triplet loss [30]. An original image $\mathbf{x}$ is used as anchor, and its reconstructions are split into positive (recognizable) and negative (unrecognizable) samples based on human annotations (Section 4). The goal is to minimize the anchor distance to positive samples and maximize that to negative ones. During inference, given an original image and its reconstruction, we use $f_\theta$ to extract their features and compute the $\ell_2$ distance between the two features.

and split its reconstructions into positive $\bar{\mathbf{x}}_i^+$ and negative $\bar{\mathbf{x}}_i^-$ samples based on human annotations. The loss function is $L = \sum_{i=1}^N \max\{d(\mathbf{x}_i, \bar{\mathbf{x}}_i^+) - d(\mathbf{x}_i, \bar{\mathbf{x}}_i^-) + \alpha, 0\}$, where $\mathbf{x}_i$ is an original image and $\bar{\mathbf{x}}_i^+$ (resp. $\bar{\mathbf{x}}_i^-$) stands for one of its recognizable (resp. unrecognizable) reconstruction, and $\alpha$ is the margin. Thus, we obtain our neural network $f_\theta$ trained on human-annotated datasets.

**Inference.** During the evaluation, $f_\theta$ is used for extracting features for original and reconstructed images. We calculate the $\ell_2$ distance between their feature vectors, that is $SemSim(\mathbf{x}, \bar{\mathbf{x}}) = \ell_2(f_\theta(\mathbf{x}), f_\theta(\bar{\mathbf{x}}))$, and then average this score over test set as the overall model performance score.

**Key Observations.** We believe SemSim captures semantic information, which plays a crucial role in privacy preservation. There are several key factors contributing to its effectiveness. (1) Being trained on human annotations enables SemSim to capture privacy leakage semantics better than metrics based on pixel-level similarity or patch CNN features. (2) By utilizing a CNN model that extracts relevant higher-level features, SemSim captures visual information related to information leakage effectively. (3) It incorporates the relationship between the original image and recognizable/unrecognizable reconstructions, improving its accuracy in assessing privacy leakage and providing better privacy assessment. SemSim has a limitation in that it requires annotated data for training. While we show that it is very generalizable and can work better than existing metrics with limited training data (refer to Figure 7), we prioritize our future work to annotate more data for even improved generalization.

## 4.4 Discussions

**Are there other ways than human perception to assess privacy leakage?** Yes. We can use a classification model trained on a dataset that contains the same categories as the reconstruction set to classify the reconstructions. If the model accurately predicts the categories, it indicates a potential privacy leakage. In our preliminary study, we used two models trained on the CIFAR-100 dataset, achieving accuracies of 82% and 65% on the test set, respectively, for recognizing reconstructed images. By using their recognition accuracies as indicators of privacy leakage, we obtained Kendall's rank correlation coefficients of 0.7023 and 0.5044 with human evaluation, respectively. These results are considered acceptable. However, there are limitations to this approach. The classification model must be trained on a dataset that matches the categories of the task, and it needs to be accurate. These limitations affect the scope of this method. Nonetheless, exploring the use of classifiers to evaluate privacy risk offers an alternative viewpoint to human perception, and it merits further investigation

**Is privacy leakage on reconstructed images a binary problem?** No. We simplify this problem by binarizing it. It can be continuous, where privacy information is leaked to a greater or lesser degree, depending on various factors such as the task and the type and amount of data that is leaked.

**How to define privacy leakage on reconstructed images in other vision tasks?** The definition depends on the task context. For example, in object counting [23], privacy information can be defined as the number of objects. Therefore, for different tasks, the definition of privacy leakage should be carefully designed and accompanied by a tailored evaluation method.

**Relationship between image quality and private leakage of reconstructed images.** The relationship between the image quality of a reconstructed image and its information leakage is complex. While better image quality can indicate better reconstruction performance, it does not necessarily imply higher privacy leakage. Conversely, a reconstructed image with poor image quality can still contain private information, while an image with higher quality may preserve privacy better. Therefore, the relationship between image quality and privacy leakage is not always straightforward and

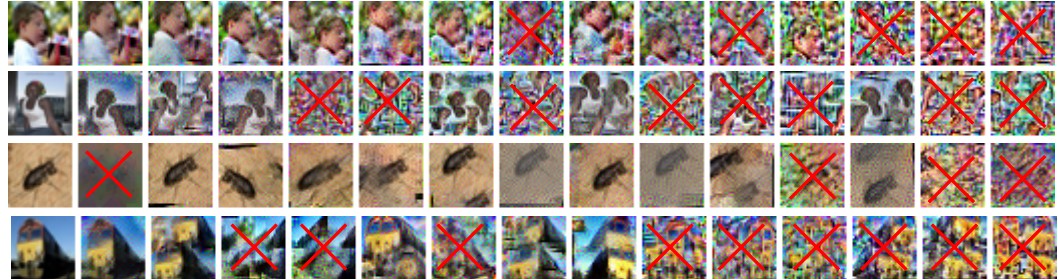

Figure 5: **Sample annotation results.** For each original image (leftmost column), its reconstructed images are placed left to right by their PSNR values from large to small. The red cross denotes that the human annotator fails to recognize the image. We observe that human evaluation is inconsistent with PSNR ranking, *e.g.*, some images that are top-ranked, or equivalently determined as high quality by PSNR are actually not recognizable by humans.

requires careful consideration and evaluation. These discussions also encourage us to explore new metrics that incorporate semantic-level information in order to better assess privacy leakage.

**Limitation and potential improvement methods for Semsim.** One limitation of SemSim is its potential performance decrease when faced with significant distributional shifts (such as using it on medical images). To address this limitation, we can annotate diverse types of data to enhance the adaptability of Semsim to a wider range of domain variations. Additionally, exploring other strategies, such as incorporating local image regions and utilizing multi-valued annotated training data, could also be considered to further enhance the effectiveness of SemSim.

## 5   Experiments

### Experimental Setups

− **Datasets.** We evaluate using the CIFAR-100 [18], Caltech-101 [5], Imagenette [2][3], CelebA [21], and Stanford Dogs [17] datasets. The first three are for generic object recognition, CelebA is for face recognition, and Stanford Dogs is a fine-grained classification dataset.
− **Classification models.** We use the following backbones: ResNet20, ResNet50, ResNet152 [9], DenseNet [12] and 8-layer CoveNet [7]. They were trained using different strategies, such as data augmentation [7], gradients with Gaussian/Laplacian noise [43], and layer-wise pruning techniques [4]. In total, there are 70 different models. Details are provided in the *supplementary material*.
− **Reconstruction attack methods.** We mainly use InvGrad [8]. In the ablation study, we evaluate SemSim using four additional attack methods, including DLG [43], CAFE [15], and GradAttack [13].
− **Correlation strength measurements.** We use two rank correlation coefficients: Spearman's rank correlation $\rho$ [31] and Kendall's rank correlation $\tau$ [16] to measure the consistency between different metrics with human perception. Values of $\rho$ and $\tau$ are between $[-1, 1]$. Being closer to -1 or 1 indicates a stronger correlation, and 0 means no correlation. For clarity, the absolute values of $\rho$ and $\tau$ are provided in the paper.

### Implementation Details

− **Classification model training.** The training of all the models to be evaluated was conducted using the PyTorch framework. The details of the classifier training, such as the specific architectures and hyperparameters used for each model, are provided in the *supplementary material*. We perform model training with one RTX-2080TI GPU and a 16-core AMD Threadripper CPU @ 3.5Ghz.
−**SemSim model training.** In the main evaluation, SemSim is trained using a learning rate of 0.1 and a batch size of 128 on the ResNet50 architecture for 200 epochs. We use leave-one-out evaluation on the 5 datasets. Some examples of the annotation data are provided in Figure 1 and Figure 5.

---

[3]Imagenette is a subset of 10 easily classified classes from ImageNet. `https://www.tensorflow.org/datasets/catalog/imagenette`

Table 1: **Comparison of different metrics on different datasets**. For each metric, we rank the 14 models and compute the correlation with rankings made by human assessment. For each test set (Column 1), SemSim is trained on the combination of the rest four datasets. Here, InvGrad [8] attack is used. $|\rho|$ and $|\tau|$ are reported. SemSim has a much stronger correlation with human annotations.

| Datasets | Metrics | PSNR | MSE | SSIM | LPIPS | FID | **SemSim** |
|---|---|---|---|---|---|---|---|
| CIFAR-100 | Spearman's $|\rho|$ | 0.6703 | 0.6176 | 0.3939 | 0.5127 | **0.7363** | **0.8637** |
| | Kendall's $|\tau|$ | 0.4725 | 0.4286 | 0.2904 | 0.3978 | **0.5604** | **0.7143** |
| Caltech-101 | Spearman's $|\rho|$ | 0.6970 | **0.7349** | 0.7218 | 0.5127 | 0.2242 | **0.8182** |
| | Kendall's $|\tau|$ | 0.5556 | **0.5525** | 0.5244 | 0.4072 | 0.1556 | **0.6889** |
| Imagenette | Spearman's $|\rho|$ | 0.5382 | 0.6395 | 0.6433 | **0.6539** | 0.4791 | **0.8257** |
| | Kendall's $|\tau|$ | 0.4349 | 0.5525 | 0.5108 | **0.5922** | 0.4252 | **0.7012** |
| CelebA | Spearman's $|\rho|$ | **0.7495** | 0.7349 | 0.6846 | 0.5824 | 0.1516 | **0.8263** |
| | Kendall's $|\tau|$ | **0.5604** | 0.5525 | 0.5264 | 0.4505 | 0.0989 | **0.6923** |
| Stanford Dogs | Spearman's $|\rho|$ | 0.4023 | 0.3968 | **0.4782** | 0.5031 | 0.3969 | **0.7120** |
| | Kendall's $|\tau|$ | 0.3537 | 0.2743 | **0.3048** | 0.3929 | 0.3196 | **0.5938** |

Table 2: **Comparison of different metrics under different attacks on the CIFAR-100 dataset**. SemSim is trained using human annotations obtained through the InvGrad [8] attack method and evaluated on different attack methods listed in the table.

| Attacks | Metrics | PSNR | MSE | SSIM | LPIPS | FID | **SemSim** |
|---|---|---|---|---|---|---|---|
| DLG [43] | Spearman's $|\rho|$ | 0.6515 | 0.6367 | 0.4069 | 0.5477 | **0.7268** | **0.8749** |
| | Kendall's $|\tau|$ | 0.4857 | 0.4174 | 0.2858 | 0.4294 | **0.5237** | **0.7342** |
| CAFE [15] | Spearman's $|\rho|$ | **0.7104** | 0.6916 | 0.5870 | 0.6793 | 0.6925 | **0.8864** |
| | Kendall's $|\tau|$ | **0.5392** | 0.4259 | 0.3318 | 0.4762 | 0.4735 | **0.7510** |
| GradAttack [13] | Spearman's $|\rho|$ | 0.6831 | 0.6944 | 0.5753 | 0.6841 | **0.7204** | **0.8437** |
| | Kendall's $|\tau|$ | 0.4943 | 0.4980 | 0.3495 | 0.4531 | **0.4819** | **0.7260** |

## 5.1  Main Evaluation

**Inconsistency between existing metrics and human perception: more results.** On each of the five test sets, we rank the 14 models according to each of the existing metrics as well as human perception. The model ranking of each metric is correlated with that from human assessment. We find that PSNR, MSE, SSIM, LPIPS, and FID do not have a high correlation with human assessment. The worst performing metric is FID: Kendal's $|\tau|$ is only 0.1556, 0.4252, 0.0989, and 0.3196 between FID and human perception, on the four test sets, respectively. While the rest four metrics exhibit a stronger correlation than FID, Kendall's $|\tau|$ is generally around 0.5, which is considered only moderate.

Moreover, from Figure 3, we find that the correlation between existing metrics themselves is often weak. For example, In Figure 3 right, Kendall's $|\tau|$ is only 0.2904 between PSNR and LPIPS. This contradiction also exists between PSNR and LPIPS and others. The above results advocate the study of new metrics that are privacy oriented.

**Comparing SemSim with existing metrics in terms of faithfulness to human perception.** We utilize SemSim to rank the models and examine its correlation with the ranking based on human perception, as shown in Table 1. We make two key observations.

First, SemSim exhibits a much stronger correlation with human perception. On the five test sets, *Kendall's $|\tau|$ is 0.7143, 0.6889, 0.7012, 0.6923, and 0.5938, respectively*, which is 0.2418, 0.1333, 0.2663, 0.1319 and 0.2401 higher than PSNR, for example. The above results suggest the risks of current metrics in the community and advocate the proposed learning-based, privacy-oriented metric.

Second, on Stanford Dogs, while SemSim is still much more faithful to human perception than other metrics, the overall correlation is lower than other datasets. Because dog species are hard to recognize, more noise was introduced to human annotation and thus to the ranking results and correlation. We speculate that fine-grained datasets are harder for privacy interception through reconstruction: humans themselves will find it hard to recognize the private content.

**Generalization ability of SemSim.** In Table 1, we adopt a leave-one-out setup, where SemSim is trained on four datasets and tested on the fifth dataset. Moreover, for each dataset, the model

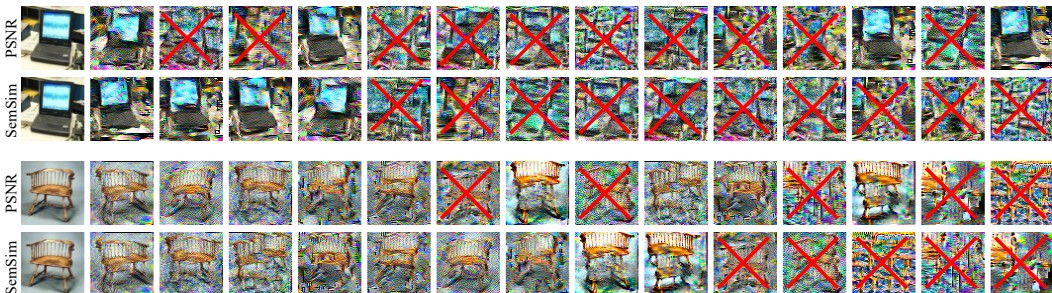

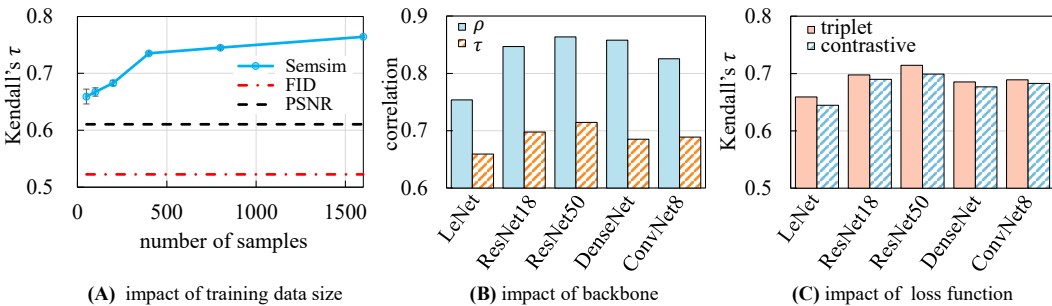

Figure 6: **Comparing the ranking of reconstruction images using PSNR and SemSim.** From the visualizations, we can observe that PSNR exhibits some inconsistencies with human perception, while SemSim consistently aligns with the judgments of human annotators. In the two examples, SemSim correctly ranks all the images with noticeable information leakage (including a laptop or chair) before the ones without or with less information leakage (that are unrecognizable). However, the rankings provided by PSNR are inaccurate for some images.

**(A)** impact of training data size    **(B)** impact of backbone    **(C)** impact of loss function

Figure 7: **Analysis of SemSim.** Evaluating the impact of (A) size of human-annotated training data, (B) variations of SemSim backbones, and (C) different loss functions for SemSim training. Experiments are conducted on the CIFAR-100 dataset.

architectures are different. For example, when using CelebA as a test set, the tested target models are ResNet50 and DesNet *etc*, while target models in training are Resnet20, 8-layer CoveNet and ResNet152 *etc*. As such, the superior results in Table 1 demonstrate the generalization ability of SemSim for test sets and model architectures.

Furthermore, we use SemSim to evaluate model vulnerability to unseen attacks. Results are provided in Table 2, where SemSim is trained using human annotations obtained through the InvGrad [8] attack method and evaluated on other attack methods such as DLG, CAFE, and GradAttack. Remarkably, we consistently observe higher correlation between SemSim and human perception compared to existing metrics. On the CIFAR-100 dataset, we observe significant improvements in Kendall's $|\tau|$ of 0.7342, 0.7510, and 0.7260, respectively, for DLG, CAFE, and GradAttack. These findings demonstrate the robustness of SemSim in capturing the privacy leakage of reconstructed images across different reconstruction attacks.

**Visualization results of SemSim.** Figure 6 presents two examples of ranking reconstruction images using PSNR and SemSim. In both cases, SemSim outperforms PSNR and provides better results.

## 5.2 Further Analysis

**Impact of the number of human annotations on SemSim training.** To evaluate this impact, we use the CIFAR100 dataset for testing and randomly select human-annotated training samples from the rest four datasets to train SemSim. Results are shown in Figure 7 **(A)**. We observe a correlation drop between SemSim and human perception as the number of training samples decreases. However, even with as few as 50 training samples (each samples includes 14 reconstructed images), SemSim outperforms existing metrics like PSNR and FID.

**Impact of different backbones for SemSim.** As mentioned in the implementation details, SemSim uses a simple ResNet50 network. Here, we try several different opinions such as LeNet and ResNet18, and present their correlation with human perception in Figure 7 **(B)**. We show that even a simple LeNet model can achieve $|\tau|$ scores higher than 0.65, surpassing the best score of 0.5604 obtained by FID. Moreover, we observe that there is a correlation between the complexity of the backbone architectures and the performance of SemSim. This indicates that utilizing more advanced and sophisticated backbone models may be able to further enhance SemSim to capture and represent visual information, leading to improved evaluation of privacy leakage in reconstructed images.

**Impact of other loss functions for SemSim training.** We further experiment with different loss functions and hyperparameters, including the contrastive loss and the triplet loss (where we set $\alpha = 1$ in experiments). From the results shown in Figure 7 **(C)**, we observe that the triplet loss shows a comparable correlation strength to the contrastive loss in relation to human assessment.

## 6  Conclusion

This paper investigates the suitability of existing evaluation metrics when privacy is leaked by a reconstruction attack. We first collect comprehensive human perception annotations on whether a reconstructed image leaks information from the original image. We find that model vulnerability to such attacks measured by existing metrics such as PSNR has a relatively weak correlation with human perception, which poses a potential risk to the community. We then propose SemSim trained on human annotations to address this problem. On five test sets, we show that SemSim has much stronger faithfulness to human perception than existing metrics. Such faithfulness remains strong when SemSim is used for different model architectures, test categories, and attack methods, thus validating its effectiveness. In future work, we will collect human perception labels from a wider source of datasets and train a more generalizable metric for privacy leakage assessment.

## Acknowledgement

This research was funded by Sony AI. We also gratefully acknowledge partial support from the ARC Discovery Early Career Researcher Award (DE200101283) awarded to Liang Zheng, the ARC Discovery Project (DP210102801), and the ARC Discovery Grant (Grant ID: DP220100800) awarded to Hongdong Li.

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

## A    Data Annotation

In **Section 4.1**, we briefly introduced how humans annotate the reconstructed images for different datasets. In the supplementary material, we have included a graphical user interface (GUI) that was utilized by the annotators. Figure 8 displays the GUI, where **(A)** and **(B)** were specifically designed for annotating different datasets.

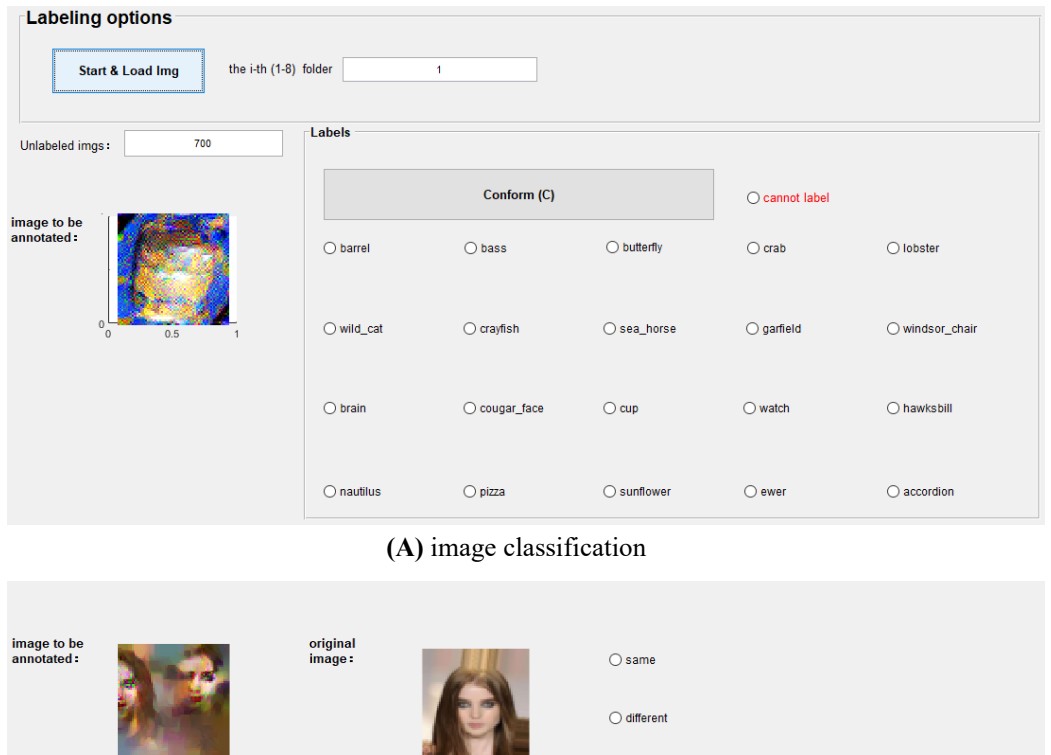

**(A)** image classification

**(B)** face recognition and fine-grained image classification

Figure 8: **Graphical user interface (GUI) used in our human annodation process.** For (A) image classification, we ask annotators to give a category to the reconstructed image. In (B) face recognition and fine-grained classification, we ask annotators to tell whether the original image and its reconstruction have the same or different identity / category.

To minimize the influence of subjective bias, we use a relatively objective formulation: whether the reconstructed image can be correctly labeled. Specifically, for CIFAR-100, Caltech-101, and Imagenette, we provide up to 20 candidate categories and see if the annotators can correctly recognize the reconstructed image; for more difficult tasks like face recognition and fine-grained classification (Celeb-A and Stanford Dogs), we give both the original and the reconstructed images and ask the annotator if they are of the same identity or species.

## B    Impact of margin value $\alpha$ in the triplet loss on SemSim

The effect of the margin parameter $\alpha$ in the triplet loss on the performance of SemSim is depicted in Figure 9. It can be observed that when $\alpha$ is set to a value close to 1, both Spearman's rank correlation ($\rho$) and Kendall's rank correlation ($\tau$) coefficients yield better results compared to other values, on CIFAR-100 and Caltech-101 datasets. We think there are two potential reasons for this observation. Firstly, if the value of $\alpha$ is too small, the model may struggle to effectively learn the discriminative features that distinguish positive (recognizable reconstructed images) and negative (unrecognizable reconstructed images) samples. On the other hand, if $\alpha$ is set to a value that is too large, the model may become excessively confident in distinguishing between positive and negative samples. However, this can lead to convergence challenges, as the loss function may have difficulty approaching 0.

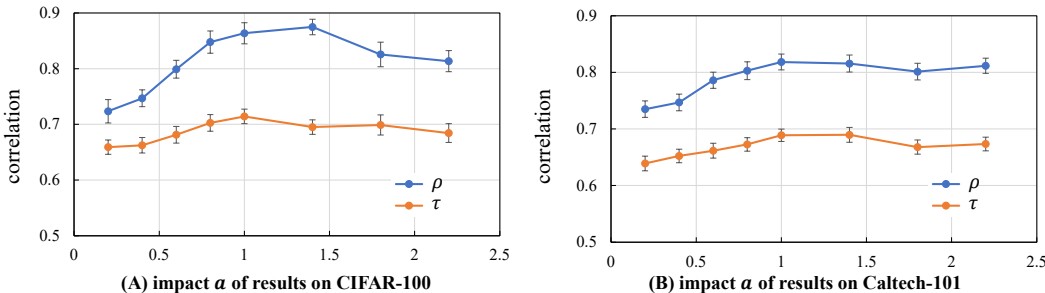

Figure 9: **Impact of $\alpha$ on SemSim when testing on (A) CIFAR-100 and (B) Caltech-101**. The margin value $\alpha$ is used in the triplet loss to ensure that negative samples are kept far apart. When evaluating the reconstructed images of CIFAR-100 or Caltech-101, we trained the ResNet50 model on the four datasets (excluding CIFAR-100 or Caltech-101) using the triplet loss. The training process involved utilizing different values of the margin parameter $\alpha$ for each dataset.

Table 3: **Details of classification models**. On each test set, we have two backbones trained without (vanilla, $2^{nd}$ column ) and with different strategies, such as using data augmentation ($3^{rd} - 6^{th}$ columns) and existing defence methods ($7^{th} - 8^{th}$ columns).

| Datasets | | Models | | | | | |
|---|---|---|---|---|---|---|---|
| CIFAR-100 | ResNet20 CoveNet8 | + Random-ResizedCrop & Random-HorizontalFlip | + TranslateX & Invert & ranslateY | + ranslateY & Autocontrast & Autocontrast | + [7] | + defense Gaussian ($10^{-3}$) | + defense Pruning (70%) |
| Caltech-101 | ResNet20 DenseNet | | | | | | |
| Imagenette | ResNet50 ResNet152 | | | | | | |
| CelebA | ResNet20 DenseNet | | | | | | |
| Stanford Dogs | ResNet50 ResNet152 | | | | | | |

In our experiments, we set $\alpha$ to 1. However, we acknowledge that there is potential for improved performance by carefully selecting the optimal value of $\alpha$ for different datasets.

## C   Classification models and training details

We conducted experiments using five datasets, CIFAR-100 [18], Caltech101 [5], CelebA [21], ImageNette [2], and Stanford dogs [17]. In our evaluation process, we considered 14 classification models for each set. Table 3 provides detailed information about these models. They were trained using stochastic gradient descent (SGD) as the optimizer, with a learning rate of 0.1.

## D   Metrics for reconstruction quality

**Mean squared error**. Assuming $x, \bar{x} \in \mathbb{R}^{n \times m}$ are two images to compare, the mean squared error (MSE) is given by,

$$\text{MSE}(x, \bar{x}) := (1/mn) \sum_{i=1}^{m} \sum_{j=1}^{n} (x_{ij} - \bar{x}_{ij})^2. \tag{3}$$

The value of MSE is between 0 and $+\infty$. The lower MSE is, the closer two images are.

**Peak-Signal-to-Noise ratio**. The Peak-Signal-to-Noise ratio (PSNR) is widely used in image quality assessment, which measures the ratio between the maximal power of a signal and its noise. Its value, expressed in dB, is given by:

$$\text{PSNR}(x, \bar{x}) = 20 \log_{10} \left( \frac{MAX_x}{\sqrt{\text{MSE}(x, \bar{x})}} \right), \tag{4}$$

where $MAX_x$ is the maximal value in the image $x$ (often replaced by 255 for `int8` images).

**SSIM**. Unlike PSNR, the structural similarity index measure (SSIM) [35] is a perception-based metric as it was designed to take into account characteristics of the human vision system through three metrics: luminance, contrast and structure of the image. It is shown that there is an analytical link between PSNR and SSIM and that it is often possible to predict one from the other for controlled perturbations (Gaussian blur, additive Gaussian noise and jpeg compressions) [11]. The above three metrics compute a pixel-wise distance between both images which is very limited when we assess semantic content of an image such as privacy leakage.

**LPIPS**. LPIPS [39], which stands for learned perceptual image patch similarity, is a perceptual metric based on a neural network aiming at correlating better with perceptual judgments. The authors take inspiration from neuroscience findings, where their model compares the activations between two images as neurons in a human cortex would. As explained in its `torchmetrics` documentation[4], a low LPIPS score indicates high similarity. Thus, in the context of privacy assessment, low LPIPS values for an original image and its reconstruction suggest high privacy leakage [13].

**Fréchet Inception Distance**. Aside from LPIPS and the aforementioned hand-crafted metrics calculated at the image level, our works also uses Fréchet inception distance (FID) [10] to measure information leakage. FID is commonly used to evaluate the domain gap between two distributions, where higher values suggest a larger domain gap. For example, FID is extensively used to evaluate the quality of images generated by generative adversarial networks (GANs) [10], by computing the distribution difference between real and generated images. In this paper, FID may reflect the difference between the original and reconstructed image distributions to reflect privacy leakage. As opposed to the pointwise metrics, FID is computed directly on image sets: $\text{InfoLeak}(\mathcal{X}, \bar{\mathcal{X}}) \propto \text{FID}(\mathcal{X}, \bar{\mathcal{X}})$.

As defined in [10, 3], given two Gaussian distributions with mean and covariance $(\boldsymbol{m}, \boldsymbol{C})$, resp. $(\bar{\boldsymbol{m}}, \bar{\boldsymbol{C}})$, FID is given by:

$$\text{FID}((\boldsymbol{m}, \boldsymbol{C}), (\bar{\boldsymbol{m}}, \bar{\boldsymbol{C}})) = \|\boldsymbol{m} - \bar{\boldsymbol{m}}\|_2^2 + \text{Tr}\left(\boldsymbol{C} + \bar{\boldsymbol{C}} - 2(\boldsymbol{C}\bar{\boldsymbol{C}})^{1/2}\right). \tag{5}$$

Its evaluation on finite sets $\mathcal{X}$ and $\bar{\mathcal{X}}$ follows verbatim by computing their empirical mean and covariance matrix. The value of FID is between $0$ and $+\infty$. The lower the FID value is, the closer two distributions are.

**Relationship between $\ell_2$ and cosine similarity**.

The $\ell_2$ norm can be used as a tool to measure the distance between vectors, often embeddings of images like those produced by our SimSem model:

$$\ell_2(\mathbf{u}, \mathbf{v}) = \|\mathbf{u} - \mathbf{v}\|_2, \tag{6}$$

The issue with distances to estimate the similarity between vectors is that they are only bounded below by zero (when $\mathbf{u} = \mathbf{v}$). This makes it hard to set a threshold above which vectors $\mathbf{u}$ and $\mathbf{v}$ can be considered dissimilar. Thus, cosine similarity is often preferred to $\ell_2$ distance as its values belong to the interval $[-1, 1]$, $1$ indicating proportional vectors and $-1$ vectors of opposite directions. Let $\mathbf{u}, \mathbf{v}$ be normalized vectors, then the relationship between cosine similarity and $\ell_2$ norm is

$$\ell_2(\mathbf{u}, \mathbf{v}) = \sqrt{2(1 - \text{cossim}(\mathbf{u}, \mathbf{v}))}. \tag{7}$$

# E   More Discussions

**How much extra effort needs to be paid to extend the current approach?** The effort required to extend the current method to other tasks also depends on the nature of the tasks. If we evaluate privacy leakage for the counting task, it would be useful to ask human annotators to count the number of objects in the reconstructed image: if it equals the number of objects in the original image, then privacy may be considered leaked. For this particular counting task, we speculate that manageable efforts will be needed to extend our current approach. On the other hand, tasks that need specialized or professional annotations will likely require more effort, such as medical image understanding.

---

[4]`https://torchmetrics.readthedocs.io/en/stable/image/learned_perceptual_image_`
`patch_similarity.html`

