# OpenReview forum: "Privacy Assessment on Reconstructed Images: Are Existing Evaluation Metrics Faithful to Human Perception?"
_NeurIPS.cc/2023/Conference — NeurIPS 2023 spotlight_

### Official Review · Reviewer_Lywc · 2023-07-01

**Soundness:** 3 good
**Presentation:** 3 good
**Contribution:** 3 good
**Rating:** 6
**Confidence:** 4

**Summary:**

The paper points out that current metrics, such as PSNR, MSE, SSIM, FID, and LPIPS, do not always reflect human's judgements on evaluation of model privacy risk under reconstruction attacks. Therefore, it proposes a learning-based measure called SemSim which is more compatible with human opinions. SemSim was trained on a binary-annotated dataset using a standard triplet loss and demonstrated good generalizability in terms of model and dataset.

**Strengths:**

The paper has good motivation and is well-written. It is the first in the literature to compare human and machine metrics for measuring privacy leakages under reconstruction attacks. The proposed measure had good generalizability.

**Weaknesses:**

This paper is based on the assumption that human perception is superior to measuring model privacy preservation ability than the machine.

The number of human annotators is small and their scores are binary instead of multi-value, e.g., from 0 to 5. The reviewer believes that using non-binary labels for a more advanced triplets loss function during training SemSim will result in better performance of the proposed metric.

It is easy to guess that a metric network trained using human scores will produce scores close to human scores.


**Questions:**

1. There is an inconsistency between the claim mentioned between lines 63-65 and the experimental setting mentioned from lines 230 to 231 and 256 to 257. I think the latter is correct, right?

2. Why does the paper use binary annotations for privacy leakage, which is less informative than multiscale ones? Furthermore, using non-binary annotations enable the usage of semi-hard triplets, improving the metric network.

3. Are there any reasons not to use the absolute values of Spearman's and Kendall's correlation? Mixing positive and negative values in the tables may cause readers confusion.

4. Is it possible to use the targeted network or something similar for measurement besides the standard metrics mentioned in the paper? For example, in face recognition, a face recognition system can be used as a metric to judge how similar the two facial images are. Another example is ImageNet classification. We can use an image classifier as a feature extractor, then calculate the distance between the embeddings of the original and the reconstructed image. Or maybe the perceptual loss? The reviewer believes that such scores should be investigated.


**Limitations:**

The authors adequately addressed the limitations.

---

> ### Author Rebuttal · Authors · 2023-08-09
>
> Thank you for your thoughtful review and valuable feedback. Below we address your concerns. Please let us know if further clarification is needed.
>
> **Q1: Inconsistency between lines 63-65 and the experimental setting mentioned from lines 230 to 231 and 256 to 257. I think the latter is correct, right?**
>
> Thank you very much for pointing out this inconsistency. Our experimental setup aligns with the description in Lines 230-231 and Lines 256-257. During rebuttal, we conducted experiments using the setup described in lines 63-65. Please refer to our reply to Q2/reviewer **j2WQ** for details. We will correct lines 63-65 for consistency in the revised paper.
>
> **Q2: Why does using binary annotations for privacy leakage which is less informative than multiscale ones?  (using non-binary annotations enable the usage of semi-hard triplets, improving the metric network.)**
>
> Great question. In this work, we use binary annotations mainly for their simplicity: a simple yet effective neural network can be trained with the standard triplet loss, providing a clear and practical methodology. As discussed in lines 194-196, privacy leakage can take continuous values or multiple scales, which has the potential to offer richer supervision for training SemSim. That said, the multi-level scoring might be limited by its subjectivity, where it is non-trivial to design a robust and consistent scoring procedure. We would like to explore this idea in our future investigation.
>
> **Q3: Are there any reasons not to use absolute values of Spearman's and Kendall's correlation?**
>
> Great suggestion. We confirm absolute values of Spearman's and Kendall's correlation can be used without causing confusion. We will use these absolute values in the revised paper for easier result interpretation.
>
>
> **Q4: Is it possible to use the targeted network or something similar for measurement besides the standard metrics mentioned in the paper? For example, in face recognition, a face recognition system can be used as a metric to judge how similar the two facial images are. Another example is ImageNet classification. We can use an image classifier as a feature extractor, then calculate the distance between the embeddings of the original and the reconstructed image. Or maybe the perceptual loss? The reviewer believes that such scores should be investigated.**
>
> *Using the target network.*  Yes, it is possible. We discussed this option in detail in Lines 183-193. While this method seems promising and easy to implement, it suffers from a few limitations.  First, it requires the original classifier to be trained on a dataset that aligns with the categories of the task at hand. Second, the accuracy of the original classifier significantly impacts the applicability of this approach, thereby constraining its scope and effectiveness. In comparison, our method can be used in a wider scope, is not dependent on classifier performance, and is demonstrated to have a good correlation with human perception.
>
> In another consideration, using the target network for privacy leakage evaluation is an alternative to human perception (Lines 183-193). Both give insights into privacy leakage, and we choose to focus on the latter in this paper.
>
> *Using the ImageNet-trained classifier for feature extraction.* Following this suggestion, we computed the ImageNet features for original and reconstructed images during rebuttal and used their feature distance to measure privacy leakage. The resulting absolute Spearman’s ρ values are 0.7495 and 0.4637 on the CIFAR-100 and Caltech-101 datasets, respectively, which are significantly lower than SemSim's 0.8637 and 0.8182. We will incorporate these discussions in the paper and explore more possible methods in future work.
>
> **Q5: This paper is based on the assumption that human perception is superior in measuring model privacy preservation ability to the machine.**
>
> Great comment. In Lines 183-193, we discuss in length the use of a classifier (machine) for privacy leakage evaluation and recognize that it can be an alternative to human opinions (Lines 192-193). In fact, we do not intend to compare human and machine in privacy leakage evaluation. We simply focus on the former, which should be an important opinion for privacy leakage and forms a nice benchmark to assess the alignment of metrics with human.
>
> **Q6: It is easy to guess that a metric network trained using human scores will produce scores close to human scores.**
>
> We agree. But please note that human scores are an important source of judging privacy leakage, so we feel it makes sense to train a metric that resembles human opinions. Importantly, through this training scheme, we find that the resulting SemSim metric has very strong generalization ability across datasets, which is critical for its future use.

---

> > ### Comment · Reviewer_Lywc · 2023-08-18
> >
> > I would like to thank the authors for their responses, which address my concerns.
> > I am looking forward to the following up work after NeurIPS.

---

### Official Review · Reviewer_j2WQ · 2023-07-02

**Soundness:** 4 excellent
**Presentation:** 3 good
**Contribution:** 3 good
**Rating:** 7
**Confidence:** 4

**Summary:**

This paper evaluates the limitations of existing hand-crafted image quality metrics in indicating privacy leakage in reconstructed images. The findings reveal weak correlation and contradictions among these metrics and with human perception, highlighting the risks of solely relying on them. To address this issue, the authors propose SemSim, a learning-based measure that assesses semantic similarity between original and reconstructed images. SemSim shows a higher correlation with human judgment compared to existing metrics when evaluating privacy leakage risk across various models, datasets, and attack methods. The contributions of this paper include valuable insights for privacy leakage assessment and the introduction of SemSim as a more reliable metric.

**Strengths:**


-	Comprehensive study: The paper conducts a comprehensive evaluation of hand-crafted image quality metrics in privacy leakage assessment across multiple datasets and models, providing valuable insights into privacy assessment for reconstructed images.

-	Clear problem statement: The paper effectively shows the weak correlation between existing metrics and human perception of privacy leakage, highlighting the significance of the problem in privacy assessment.

-	Proposed learning-based measure: SemSim, which considers the semantic similarity between the original and rebuilt images, is useful for a more precise assessment of privacy leakage. In comparison to other metrics, SemSim shows a higher association with human judgement when assessing privacy leaks.

-	Good results: The authors demonstrate the potential generalizability of SemSim's correlation with human judgment on different datasets.

-	New datasets: This paper provides new datasets for further research in privacy assessment.

-	Insightful Discussion: The paper provides insightful discussions on privacy leakage in reconstructed images.


**Weaknesses:**


-	Lack of details about existing metric FID: Providing more information about the specific models used for feature extraction in the FID metric can enhance reproducibility and clarity.

-	Domain generalization ability of the proposed metric should be discussed, such as only having one set of reconstructed images for training. It is important for the application of proposed measure in real world.

-	L2 distance is used by Semsim. What will happen to Semsim's performance if alternative distance measures are used? Ablation study should be provided.

-	Typos: The paper has a few typos, such as in line 241 with the value "0-.0989," which should be corrected.


**Questions:**

Please address the above weaknesses.

**Limitations:**

The authors have discussed the limitations, e.g. the requirement of training data.

---

> ### Author Rebuttal · Authors · 2023-08-09
>
> Thank you for your thoughtful review and valuable feedback. Below we address your concerns. Please let us know if further clarification is needed.
>
> **Q1: Details of FID.**
>
> In lines 118-120 of the submission, we referred to [8] where FID is utilized to assess the similarity between generated and real images. Its calculation uses Eq. 3 in the Supplementary Material. Specifically, we use FID to compute the distance between a set of reconstructed images and a set of original images. Specifically, we use InceptionV3 pre-trained on ImageNet to extract features from both image sets, resulting in two sets of feature vectors. Then, we compute the mean and variance for each set and use Eq. 3 in the Supplementary Material to calculate FID between the reconstructed and original images. We will provide a more detailed introduction to FID computation in the Supplementary Material.
>
>
> **Q2: Domain generalization ability of the proposed metric should be discussed, such as only having one set of reconstructed images for training.**
>
> We greatly appreciate this suggestion and will add more discussion in the revised paper. In all of our experiments, we apply SemSim on datasets with (mostly) disjoint label spaces from datasets where SemSim is trained. We show that SemSim exhibits higher faithfulness to human perception than existing metrics. Notably, in Section 5.1, SemSim surpasses FID and PSNR on CIFAR-100, even when it is trained with only 50 human annotations from datasets other than CIFAR-100. It illustrates the capacity of SemSim to generalize to unseen data with limited training data.
>
> During rebuttal, following the suggestion, we trained SemSim on one dataset (CIFAR-100) and applied it to other datasets than CIFAR-100. Results in the table below indicate that SemSim achieves higher absolute Spearman correlation values on all four test sets, compared with existing metrics. This further demonstrates the generalization capability of SemSim and will be added to the revised paper.
>
>
> | Dataset | Abs(Spearman’s $\rho$) | SemSim|
> |--------|--------|--------|
> | Caltech-101  | 0.7349 (MSE)   |  0.7517  |
> | Imagenette   | 0.7349 (LPIPS)  | 0.7653  |
> | CelebA       | 0.7495  (PSNR)  | 0.7682  |
> | Stanford Dogs| 0.5031 (LPIPS)   | 0.6194  |
>
> Limitation. We recognize that the effectiveness of SemSim may be influenced when being used for very different tasks from its training domains such as medical images. In Lines 177-181, we discuss potential strategies to improve SemSim's flexibility to broader domain variations through annotating diverse data types, which we will investigate in future work. These discussions will be incorporated into the revised text.
>
>
> **Q3: L2 distance is used by SemSim. What will happen to SemSim's performance if alternative distance measures are used?**
>
> **L2 distance is also used in the triplet loss for SemSim training**, so using L2 distance during inference is a natural choice. Moreover, during both training and inference, L2 distance is computed on the **normalized feature vectors** of reconstructed and original images. So if we use cosine similarity, there will be no change to the correlation observations. We will include this discussion in the revised paper.
>
>
> **Q4: Typo.**
>
> Thank you for pointing out the typos. We will correct these typos and double check our paper to ensure accuracy and clarity.

---

> > ### Comment · Reviewer_j2WQ · 2023-08-16
> >
> > Thanks for the response. It solved my concerns on the genealization performance.

---

### Official Review · Reviewer_oRwe · 2023-07-07

**Soundness:** 4 excellent
**Presentation:** 3 good
**Contribution:** 4 excellent
**Rating:** 8
**Confidence:** 5

**Summary:**

The paper establish a system for human to annotate whether a reconstruction (from privacy attacks) is recognizable (thus breaches privacy). Based on human annotation results, a new learning-based metric is proposed, which shows significantly better alignment and faithfulness with human perception.

**Strengths:**

1. High originality and novelty. To my knowledge, the work is the first to utilize human annotation to address the faithfulness of evaluation metrics in privacy attacks.
2. Meaningful for the machine learning privacy community. In terms of breaching privacy, it is highly important and meaningful to align the results with human evaluations. This area currently lacks such faithful and aligned metrics to evaluate whether a privacy attack is indeed successful.
3. Paper written in good quality: motivation is well-depicted, the approach is described in good clarity. The figures give clear and convincing illustrations.

**Weaknesses:**

No obvious weakness, good work.

**Questions:**

1. The authors claim that the metric generalizes well among several datasets. I am curious whether it can still generalize well enough when the distribution change a lot, for example, on medical imaging datasets (However as you said, the privacy definition of such tasks could be different from a professional view, for example, a doctor may no longer think patch-level similarity is important in privacy for medical images). Readers could be happy to see more discussions on how the definition of "privacy" could change for other cases, and figure out how many extra efforts need to be paid to extend the current approach (For example changing ordinary human labelers to professionals like physicians)

**Limitations:**

1. If we want to make the evaluation metric more robust and more generalizable, in the future, we may consider scaling up the annotations to more images and more datasets. Even if we don't scale up, for new tasks or new datasets, we may need to go through the annoation from scratch. In other words, the scalability of the system is somewhat limited.

2. The metric is designed to evaluate reconstruction similarity. However, if such an evaluation metric is open-sourced and accessible to everyone, attackers may use this metric to train an even stronger privacy attack model. The risk needs to be further discussed and carefully handled.

---

> ### Author Rebuttal · Authors · 2023-08-09
>
> We would like to thank the reviewer for the constructive comments. We address the concerns raised by the reviewer below. Please let us know if further clarification is needed.
>
> **Q1: Can the proposed method generalize well enough when the distribution changed a lot, such as medical images?**
>
> Great question. The proposed method is shown to perform well on a diverse range of tasks such as generic object recognition, face recognition, and fine-grained classification. For medical images, it is not feasible for us to conduct experiment in a short time to verify the effectiveness of SemSim, because expert annotations are needed to judge whether a reconstructed image still exhibit certain medical conditions. In our best guess, SemSim would be less effective on medical images due to the very different styles (*e.g.,* X-rays and ultrasound image). We discussed potential strategies to improve SemSim's flexibility to broader domain variations through annotating diverse data types (please refer to Lines 177-181), which we will investigate in future work. These discussions will be incorporated into the revised paper.
>
> **Q2: Provide more discussion on 1) how the definition of "privacy" could change for other cases, and 2) how much extra efforts need to be paid to extend the current approach *e.g.,* considering professionals for annotation**
>
> Thank you for raising these points. In Lines 197-200, we discussed that the definition of privacy may vary in different tasks. For example, for object counting, privacy information may reside in the number of objects instead of individual objects themselves. We will enhance this section by adding insights from our reply to Q1.
>
> The efforts required to extend the current method to other tasks also depend on the nature of the tasks. If we evaluate privacy leakage for the counting task, it would be useful to ask human annotators to count the number of objects in the reconstructed image: if it equals the number of objects in the original image, then privacy may be considered leaked. For this particular counting task, we speculate that manageable efforts will be needed to extend our current approach. On the other hand, tasks that need specialized or professional annotations will likely require more efforts, such as medical image understanding. These discussions will be integrated into the paper.
>
> **Q3: Scaling up the annotations to more images and datasets.**
>
> Great suggestion. In Lines 179-181 of the submission, we recognize the importance of scaling up data annotation to more images and datasets and will do so in our future investigation.
>
> **Q4: Risk of open source code.**
>
> Interesting comment. By open-sourcing the metric and training data, we intend to promote transparency and collaboration. We agree on the mentioned risk. But we would like to point out that SemSim will also allow for better development of defence mechanisms and privacy-preserving models. In light of this comment, we will consider releasing SemSim under licenses that only allow for academic uses.

---

### Official Review · Reviewer_AN5f · 2023-07-13

**Soundness:** 3 good
**Presentation:** 3 good
**Contribution:** 2 fair
**Rating:** 6
**Confidence:** 3

**Summary:**

This paper studies the faithfulness of hand-crafted metrics to human perception of privacy information from the reconstructed images, and discovers hand-crafted metrics only have a weak correlation with the human evaluation of privacy leakage. A learning-based measure called SemSim is proposed to evaluate the Semantic Similarity between the original and reconstructed images.

**Strengths:**

+ a comprehensive comparison of traditional metrics for privacy leakage of reconstructed image
+ proposes a learning-based metric with human ratings
+ improves over all existing image quality metrics


**Weaknesses:**

- The definition of privacy is based on image content classification, which may miss important privacy attributes in local image regions.
- The details of human rating collection procedure are needed to ensure no subjective bias in the metric training. For example, who are the people giving the rating, how many participants, what questions are asked to the participant.
- The metric learning method is very simple.


**Questions:**

- Can we just apply the original classifier on the reconstructed images as a metric for privacy leakage?
- According to Table 1, how to explain PSNR/MSE is sometimes better than LPIPS/FID, which is used more widely to measure image generation quality?


**Limitations:**

Important details missing for human study, which is centric to the learned metric.

---

> ### Author Rebuttal · Authors · 2023-08-09
>
> We thank the reviewer for many insightful comments. We answer the questions in what follows. Please let us know if further clarification is needed.
>
> **Q1: Can we just apply the original classifier on the reconstructed images as a metric for privacy leakage?**
>
> Yes, and we have discussed this possible method in Lines 183-193. This method has some limitations. First, it requires the classifier to be trained on a dataset that has the same categories with the task at hand. Second, the classifier accuracy significantly impacts the effectiveness of this approach. These limitations constrain the scope and effectiveness of using the original classifier for privacy leakage measurement. We will add more discussions about this method to highlight its potential benefits and inherent constraints in the revised paper.
>
>
> **Q2: According to Table 1, how to explain PSNR/MSE is sometimes better than LPIPS/FID, which is used more widely to measure image generation quality?**
>
> Great question. Table 1 is on privacy leakage assessment, which is different from measuring image generation quality. In Lines 201-208 we discussed the connections and differences between these two evaluation problems. Due to the difference, although LPIPS and FID are widely used to **measure image generation quality**, they are not necessarily better than PSNR/MSE in **measuring privacy leakage**. Some visual examples of using LPIPS, PSNR and MSE for privacy leakage assessment are provided in Fig. 1 in the submission: in some cases LPIPS is consistent with human opinion while in others PSNR/MSE is more aligned.
>
> **Q3: The definition of privacy is based on image content classification, which may miss important privacy attributes in local image regions.**
>
> We agree that image content classification may miss important privacy attributes in local regions. It would be an interesting direction for improvement, along with a few other suggested ways (please refer to our reply in "general response to all reviewers" for more discussions).
>
> Nevertheless, as an early attempt to define privacy using image semantics, our current definition based on global image content has proven to be more faithful to human perception than existing metrics in evaluating privacy leakage.  We will add these discussions to our paper.
>
>
> **Q4: The details of human rating collection procedure are needed to ensure no subjective bias in the metric training.**
>
> Thanks for the kind suggestion. In Section 4.1, we detailed the number of participants and the annotation procedure. Specifically, each image or image pair is annotated by 5 independent annotators randomly selected from a pool of 23 annotators. These annotators are from various backgrounds, including college students, housewives, and part-time workers from different industries.
>
> To further minimize the influence of subjective bias, we use a relatively objective formulation: whether the reconstructed image can be correctly labeled. Specifically, for CIFAR-100, Caltech-101, and Imagenette, we provide up to 20 candidate categories and see if the annotators can correctly recognize the reconstructed image; for more difficult tasks like face recognition and fine-grained classification (Celeb-A and Stanford Dogs), we give both the original and the reconstructed images and ask the annotator if they are of the same identity or species.
>
> Detailed graphical user interfaces (GUI) for annotation are shown in Section A of the Supplementary Material.
>
> We will add these details in the revised text.
>
> **Q5: The metric learning method is very simple.**
>
> We agree that the proposed metric is simple and feel that being simple is an advantage. In the comprehensive evaluation, this simple metric outperforms existing metrics *w.r.t* faithfulness to human perception. This conclusion would benefit future explorations in semantic-aware privacy assessment, while the metric itself will create opportunities for further refinement and study.

---

### Author Rebuttal · Authors · 2023-08-09

We thank all reviewers for the thoughtful feedback. We are greatly encouraged that the reviewers appreciate this manuscript for its originality and novelty, its implications for the machine learning privacy community (reviewer **oRwe**),  clear motivation (reviewers **oRwe** and **Lywc**), comprehensive comparisons (reviewers **AN5f** and **j2WQ**), and good writing (reviewers **oRwe** and **Lywc**).

Below we discuss a common question summarized from reviewers' comments. We will then answer the questions of each reviewer separately.

* **How to further improve the performance of the learning-based metric?**
As an initial work utilizing human annotation to address the faithfulness of evaluation metrics in privacy attacks (Reviewer **oRwe**), we acknowledge that there is room for further improvement to the proposed SemSim. As suggested by reviewers, potential solutions include using local region information (Reviewer **AN5f**), non-binary labels (Reviewer **Lywc**), and annotating more data to enlarge the application scope (Reviewer **oRwe**). In the current manuscript, we also discussed several potential ways on how to further improve SemSim, *e.g.,* Lines 180-181 and 284-256. With that said, we believe that the potential for further improvement does not impact the contribution of work, which offers valuable insights into privacy assessment and can further stimulate efforts within the community to develop more robust evaluation metrics.

We will answer the questions of ethics reviewers during the discussion session, following the guidelines for rebuttal replies.

---

### Decision · Program_Chairs · 2023-09-21

**Decision:**

Accept (spotlight)

**Comment:**

The paper received positive scores initially: 1 Strong accept, 1 accept, 2 weak accept. The following issues were identified:
1) the definitely of privacy for image content classification misses privacy in local regions. (AN5f)
2) details missing about the procedure for human rating collection (AN5f)
3) simple metric learning method (AN5f)
4) how well does the metric generalize? what is the scalability? (oRwe, j2WQ)
5) opening up the metric may allow attackers to build stronger attack models (oRwe)
6) missing ablation study on the metric used (j2WQ)
7) would non-binary labels be better? (Lywc)
8) are other similarity measurements possible for specific tasks? (Lywc)
9) an assumption that human perception is superior in measuring privacy preservation. (Lywc)

The authors wrote a response, and all reviewers were satisfied by the responses. The reviewers appreciated the high originality and novelty, as well as the meaningful impact to the ML privacy community, new dataset and good results, and the quality of the paper presentation.
Thus, the AC recommends accept. The authors should update the paper according to the review, response, and discussion.